# Application of Upstream Open Reading Frames (uORFs) Editing for the Development of Stress-Tolerant Crops

**DOI:** 10.3390/ijms22073743

**Published:** 2021-04-03

**Authors:** Taeyoung Um, Taehyeon Park, Jae Sung Shim, Youn Shic Kim, Gang-Seob Lee, Ik-Young Choi, Ju-Kon Kim, Jun Sung Seo, Soo Chul Park

**Affiliations:** 1Agriculture and Life Sciences Research Institute, Kangwon National University, Chuncheon 24341, Korea; tyoungum@kangwon.ac.kr (T.U.); younshic@kangwon.ac.kr (Y.S.K.); 2Crop Biotechnology Institute, GreenBio Science and Technology, Seoul National University, Pyeongchang 25354, Korea; qkrxogus95@snu.ac.kr (T.P.); jukon@snu.ac.kr (J.-K.K.); 3School of Biological Sciences and Technology, Chonnam National University, Gwangju 61186, Korea; jsshim@chonnam.ac.kr; 4Department of Agricultural Biotechnology, National Academy of Agricultural Science, Rural Development Administration, Jeonju 54874, Korea; kangslee@korea.kr; 5Department of Agricultural and Life Industry, Kangwon National University, Chuncheon 24341, Korea; choii@kangwon.ac.kr

**Keywords:** upstream open reading frames (uORFs), stress-resistant crops, gene editing, CRISPR, enhancing the gene expression

## Abstract

Global population growth and climate change are posing increasing challenges to the production of a stable crop supply using current agricultural practices. The generation of genetically modified (GM) crops has contributed to improving crop stress tolerance and productivity; however, many regulations are still in place that limit their commercialization. Recently, alternative biotechnology-based strategies, such as gene-edited (GE) crops, have been in the spotlight. Gene-editing technology, based on the clustered regularly interspaced short palindromic repeats (CRISPR) platform, has emerged as a revolutionary tool for targeted gene mutation, and has received attention as a game changer in the global biotechnology market. Here, we briefly introduce the concept of upstream open reading frames (uORFs) editing, which allows for control of the translation of downstream ORFs, and outline the potential for enhancing target gene expression by mutating uORFs. We discuss the current status of developing stress-tolerant crops, and discuss uORF targets associated with salt stress-responsive genes in rice that have already been verified by transgenic research. Finally, we overview the strategy for developing GE crops using uORF editing via the CRISPR-Cas9 system. A case is therefore made that the mutation of uORFs represents an efficient method for developing GE crops and an expansion of the scope of application of genome editing technology.

## 1. Introduction

Crop productivity is increasingly threatened by climate change, and this issue is exacerbated by the growing global population. For example, climate change is causing environmental stress, as exemplified by elevated temperatures, drought, and flooding. On the other hand, global population growth and urbanization has resulted in the loss of agriculturally viable land. According to a study of the correlation between crop yields and global warming, an increase of 1 degree Celsius in global temperatures could result in a reduction of the global production of wheat by 6.0%, rice (*Oryza sativa*) by 3.2 %, corn (*Zea mays*) by 7.4 %, and soybeans (*Glycine max*) by 3.1% [1]. Additionally, it is expected that the global population will reach almost 9 billion by 2050, and the expansion of urbanization will result in an 80% loss of agricultural land by 2030 in Asia and Africa. Consequently, an increase of more than 70% of current crop production will be required to maintain food security [2,3,4].

Genetically modified (GM) crops provide a strategy to increase stress resistance and crop productivity through genetic improvement in relatively short time frames compared to traditional breeding approaches [5]. Furthermore, their generation can enhance the understanding of molecular mechanisms that affect plant development and defense, and this information has provided opportunities to improve crop yield [6,7,8]. However, regulatory hurdles for GM crop commercialization, including safety evaluation, have necessitated the development of alternative strategies involving gene-editing (GE), such as the use of zinc-finger nucleases (ZFNs), TAL effector nucleases (TALENs), and clustered regularly interspaced short palindromic repeats (CRISPR) tools [9]. As an example, Calyxt has developed a TALEN-based genome-edited soybean with modified oil composition and commercialized the derived product, high oleic soybean oil, on the US market [10]. In contrast to the previous GM techniques, which randomly introduce changes, GE allows for precise modification and is identical to those derived from conventional breeding [11,12]. Thus, it is anticipated that many GE crops could be a more acceptable product than GM crops [13].

Messenger RNA (mRNA) is a transcript of the gene, and generally composed of an untranslated region (UTR and intron) and a translated coding region (exon). Upstream open reading frames (uORFs) are a *cis*-element in the 5’ untranslated regions (UTRs) of mRNAs that induce ribosome stalling and dissociation during the translation of mRNAs [14,15]. Most uORFs negatively regulate the expression of protein encoding main ORF (mORF) of mRNA (Figure 1A). During the translation process, the uORF in the 5’UTR is translated to short peptides, and most uORFs have a start codon (AUG), although some exceptions have been identified [14,16]. It has been predicted by sequence analysis that 49% and 44% of the total transcripts of humans and mice, respectively, have uORFs, as have 20%~40% of the total transcripts of plants such as rice, *Arabidopsis thaliana*, and corn [17,18]. Most stress-responsive transcripts harboring uORF are induced by specific environmental conditions, and it is known that mutations of uORF sequences reduce their capacity for negative regulation [19,20].

This review describes the strategic application of uORF sequences for developing salt stress-tolerant crops, focusing on the enhancement of the target gene expression by the mutation of uORF using CRISPR-Cas9 (Figure 1B), and discusses a general approach for developing GE crops using uORF editing.

## 2. Upstream Open Reading Frames (uORF)

### 2.1. The Process of Translation of mRNA Mediated by uORF

During mRNA translation, uORF translation causes ribosome stalling, leading to repressed translation of the mORF. This process can be divided into three stages: initiation, elongation, and termination [15,21,22,23,24]. First, the start codon in uORF is recognized by scanning the 40S ribosomal subunit and eukaryotic initiation factors (eIFs), then, in the second stage, the 60S and 80S ribosomal complexes bind to the 40S, allowing for the elongation of the polypeptide chain. Finally, the third stage involves the termination of translation in the uORF, which results in repressed translation of the mORF (Figure 2A). The detailed process can be described in the following steps: (1) ribosome stalling, physical interaction between the peptide encoded by the uORF and ribosomes; (2) the translational machinery dissociates from the mRNA after uORF translation and the mORF is not translated; (3) trigger mRNA decay, nonsense-mediated mRNA decay (NMD) is induced due to the presence of the stop codons in uORF and the recognition as premature transcript; and (4) decreasing mORF translation, after uORF translation elongation, the 40S and 60S dissociate from the uORF, and the 40S unit remains associated with the mRNA. Translation of the mORF is then re-initiated.

### 2.2. The Classification of uORF

The uORFs are classified by the position of the uORF stop codon [25]. Type 1: the location of the stop codon of the uORF is independent of the mORF. Type 2: the stop codon of the uORF overlaps with the mORF. Type 3: the stop codon of the uORF is the same as that of the stop codon of the mORF (Figure 2B). In rice, it was expected that uORF containing genes are 20.65% of the total genes. Type 1 and Type 2 account for 87.63% and 12.17% of the total uORFs, respectively, and there are only 41 examples of Type 3 uORFs [18,25].

### 2.3. Stress-Responsive uORF-Mediated Transcripts in Rice

Drought and the salinization of soils are accelerated by climate change and lead to a significant reduction in crop yield [26,27]. These environmental phenomena cause cellular dehydration by reducing the water content of the cytoplasm, leading to substantial physiological and cytological changes [28,29,30]. To survive under such stressful conditions, plants have developed a range of mechanisms to reduce the effects of drought stress, many of which are associated with extensive changes in gene expression [31,32]. Notably, it was reported that most stress-responsive transcripts harbor uORFs in plants [21,25]. Rice is an excellent model to study the significance of uORFs in stress-related gene expression since its genome has been extensively characterized, it is a major crop, and it is sensitive to drought and salt stress [33,34]. As part of a mechanism associated with salt stress, members of the chloride channel protein (CLC) family, which is found in prokaryotes and eukaryotes, play an important role in ion homeostasis [35,36]. A characterized example in rice is the OsCLC1 protein, which is located at the tonoplast in the cell [37], and overexpression of *OsCLC1* in rice was found to enhance drought stress tolerance through modulating jasmonic acid (JA) signaling [38]. We predicted the uORF in *OsCLC1* transcripts using uORFFlight (http://uorflight.whu.edu.cn accessed on 20 February 2021) and identified a type 1 uORF in the 5’UTR. The uORF is comprised of 33 nucleotides, including the start codon (ATG) and stop codon (TGG). We also analyzed the uORFs in the 5’UTRs of the salt stress-responsive rice genes (Table 1) through the identification of 107 salt stress-responsive genes reported in the literature, and predicted 393 uORFs in the 46 transcripts related to the transporter, transcription factor, and kinase functional categories. Of these, type 1 uORFs accounted for 91%, while only 8.7% were of type 2, and type 3 uORFs were not represented.

### 2.4. Avoidance of uORF-Mediated Repression of Gene Expression during Stress Responses 

Under stressful conditions, plants activate their defense systems and limit their growth, which is associated with resource restrictions [39,40]. The interaction between defense and growth mechanisms in plants involves in a well-documented tradeoff system. Antagonistic interactions between JA- and gibberellic acid (GA)-regulated processes are one of the cases [41,42,43]. uORF-mediated translational control represents another instance of a tradeoff system. For example, the expression of uORF-associated transcripts is down-regulated by uORF translation under normal conditions to facilitate plant growth. To upregulate their expression under stress conditions, strategies are employed to avoid uORF-mediated translation. These include leaky scanning of the uORF and the reinitiation of the translation process, and the alternative splicing and selection of the transcription start site during the transcription process [14,43,44,45,46,47]. (1) Leaky scanning of uORF: 40S ribosome subunits initiate the translation on the mRNA 5’ cap and recognize the uORF before the mORF (Figure 1A). Unlike regular scanning, 40S ribosome subunits do not scan the uORF start codon, and recognize the mORF start codon through leaky scanning. (2) Reinitiation of translation: After translation of the uORF, the 60S ribosome subunits are released, and only 40S ribosome subunits bind to the mORF (Figure 2A). When mORF translation is required, the remaining 40S ribosome subunits in mRNA recognize the mORF start codon and associate with the 60S subunits again. (3) Alternative slicing: the uORF start codon or uORF sequences in the intron are removed by the splicing of pre-mRNA. The transcript without the uORF is generated from pre-mRNA (Figure 3A). (4) Selection of the transcription start sites: the transcripts of numerous genes are determined by multiple transcription start sites (TSSs). Under normal conditions, transcripts with a uORF are produced from TSSs located upstream of the uORF, whereas under stress conditions, transcripts without a uORF are produced from TSSs located downstream of the uORF (Figure 3B).
ijms-22-03743-t001_Table 1Table 1Analysis of uORF in salt stress responsive genes.GeneGene ID (Rap-DB)Transcript ID (MSU)uORF TypeExpression ResponseReference123*HAK1*Os04g0401700LOC_Os04g32920.2100down[48]LOC_Os04g32920.4400*HAK10*Os06g0625900LOC_Os06g42030.1100up[49]*HKT4*Os04g0607500LOC_Os04g51820.1200up[49]LOC_Os04g51820.2200LOC_Os04g51820.33120*HKT6*Os02g0175000LOC_Os02g07830.1210down[49]*HKT8*Os01g0307500LOC_Os01g20160.1100down[49]*NHX1*Os07g0666900LOC_Os07g47100.1100up[50]LOC_Os07g47100.2100LOC_Os07g47100.3100*SOS1*Os12g0641100LOC_Os12g44360.1930up[50]LOC_Os12g44360.2010LOC_Os12g44360.3010LOC_Os12g44360.4010*CAX3*Os04g0653200LOC_Os04g55940.1200up[51]LOC_Os04g55940.2100*IDS1*Os03g0818800LOC_Os03g60430.1200down[52]LOC_Os03g60430.2200*NAC6*Os01g0884300LOC_Os01g66120.1100up[53]LOC_Os01g66120.2300*EIL2*Os07g0685700LOC_Os07g48630.1100down[54]LOC_Os07g48630.21200*bHLH035*Os01g0159800LOC_Os01g06640.1300up[55]LOC_Os01g06640.2200LOC_Os01g06640.31400*LEA5*Os05g0584200LOC_Os05g50710.1100up[56]*JAZ9*Os03g0180800LOC_Os03g08310.1100down[57]*RSS3*Os11g0446000LOC_Os03g08310.1100up[58]*ACA6*Os01g0939100LOC_Os01g71240.1300up[59]*RLCK3*Os01g0113300LOC_Os01g02300.1100up[60]*RLCK5*Os01g0114100LOC_Os01g02390.1010up[60]LOC_Os01g02390.2010*RLCK6*Os01g0114300LOC_Os01g02400.1000up[60]LOC_Os01g02400.21710LOC_Os01g02400.33900*RLCK27*Os01g0247500LOC_Os01g14510.1800up[60]*RLCK42*Os01g0602800LOC_Os01g41870.1100down[60]LOC_Os01g41870.2100*RLCK48*Os01g0852100LOC_Os01g63280.12030down[60]LOC_Os01g63280.2110LOC_Os01g63280.31430*RLCK57*Os01g0973500LOC_Os01g74200.1100up[60]*RLCK72*Os02g0513000LOC_Os02g30900.1200up[60]*RLCK80*Os02g0650500LOC_Os02g43430.1100up[60]*RLCK85 (ER2)*Os02g0777400LOC_Os02g53720.1100down[60]LOC_Os02g53720.2100*RLCK95*Os03g0113000LOC_Os03g02190.1010down[60]*RLCK101 (PTK3)*Os03g0159100LOC_Os03g06330.1900up[60]*RLCK106*Os03g0264300LOC_Os03g15770.1400up[60]LOC_Os03g15770.2110LOC_Os03g15770.3500*RLCK110 (PTK5)*Os03g0407900LOC_Os03g29410.1200up[60]*RLCK167*Os04g0654600LOC_Os04g56060.1300down[60]*RLCK168*Os04g0655300LOC_Os04g56110.1600up[60]LOC_Os04g56110.22300LOC_Os04g56110.3500*RLCK191*Os05g0589700LOC_Os05g51190.2100down[60]LOC_Os05g51190.3100*RLCK194 (PUB5)*Os06g0140800LOC_Os06g04880.1200up[60]*RLCK204*Os06g0203800LOC_Os06g10230.1110down[60]*RLCK216 (RRK1)*Os06g0693200LOC_Os06g47820.1100up[60]*RLCK218*Os06g0714900LOC_Os06g50100.1600up[60]*RLCK223*Os07g0159700LOC_Os07g06570.1100down[60]*RLCK242*Os07g0693000LOC_Os07g49240.1100down[60]*RLCK249 (ER1)*Os06g0203800LOC_Os06g10230.1110down[60]*RLCK301*Os10g0442800LOC_Os10g30600.1100up[60]LOC_Os10g30600.2500*RLCK334*Os11g0556400LOC_Os11g35274.1100down[60]*RLCK373*Os12g0611100LOC_Os12g41710.1400down[60]LOC_Os12g41710.23000LOC_Os12g41710.31600LOC_Os12g41710.42900*RLCK375*Os12g0615300LOC_Os12g42070.1700down[60]*Aldh2a*Os02g0730000LOC_Os02g49720.6710up[61]*ABA2*Os03g0810800LOC_Os03g59610.1100up[55]

## 3. Genetically Modified (GM) Crops

Various strategies have been used to develop biotech crops, including GM crops, which are projected to be important for meeting crop yield targets and food security goals [62,63]. The development of GM crops began in the 1980s and focused on the insertion of useful genes into plant genomes or the suppressed expression of undesirable genes through genetic engineering coupled with innovative plant transformation techniques [64,65,66]. Plant genetic engineering was further accelerated by discovery-based studies of model experimental species, such as *Arabidopsis thaliana* [67]. The alteration or introduction of agronomically important traits, such as insect resistance or herbicide tolerance, can be achieved through the insertion of a foreign gene into the plant genome. Indeed, the first generation of transgenic crops included corn, soybean, and cotton lines, into which bacterial genes were introduced to confer herbicide resistance or pest resistance [68,69,70]. The development of such crops’ tolerance has led to reductions in the application of pesticides and an expansion of weed management options [71,72].

### 3.1. Development of the First GM Crops

In 1994, the Flavr Savr^TM^ tomato became the first commercial GM crop to have food and environmental safety approval. This commercial variety harbored a native gene encoding the cell wall degrading enzyme polygalacturonase that had been inserted in the antisense direction, with the intent of delaying fruit softening, and thus extending shelf life. To date, GM crops have been developed for at least 403 varieties of 29 crops targeting multiple traits, including herbicides and pest resistance, and while the worldwide area cultivated with GM crops was only 1.7 million hectares in 1996, it reached 190.4 million hectares by 2019 [73]. This area continues to increase annually; the United States cultivates the greatest area of GM crops (71.5 million hectares), followed by Brazil (52.8 million), Argentina (24.0 million), Canada (12.5 million), and India (11.9 million). Between 1996 and 2019, herbicide tolerance and insect resistance were the principal target traits, but a more recent focus has been on the development of stacked traits, including increased yield and tolerance to abiotic and biotic stresses. The cultivation area of GM crops with stacked traits increased from 58.5 million hectares in 2015 to 85.1 million hectares in 2019 [73].

Since the development of the first GM crops, a variety of genes and hormone regulated pathways that affect plant stress response have been identified, and GM crops have been developed to improve traits such as yield and tolerance to abiotic stresses, including drought and high salinity [74]. Thus, elucidation of the molecular mechanisms of hormone biosynthesis and signaling pathways are providing new genes to target for developing GE crops [7,8].

### 3.2. Hormone Regulation in Plant Salt Stress Response

Many details of the molecular mechanisms by which the phytohormones abscisic acid (ABA) and JA regulate salt and drought stress responses have been uncovered. It has been shown that JA signal transduction is propagated by the degradation of the jasmonate ZIM domain (JAZ) repressor proteins. This degradation allows transcription factors (TFs) to bind to the promoter of JA response genes, which include salt-responsive genes, inducing their transcription [75,76]. As an example, it was reported that, in rice, OsJAZ9 regulates transcription of the TF *OsbHLH062*, which is involved in salt stress tolerance in rice [57]. ABA belongs to a class of metabolites known as isoprenoids, also called terpenoids [77], and conjugated forms of ABA play an important role in ABA homeostasis. Examples include phaseic acid, a catabolite of ABA that is generated through the action of an *ABA* 8′-hydroxylase (ABA8ox), and an inactivated glucose-conjugated form, mediated by ABA glucosyltransferase (AOG). Accordingly, loss of function mutants of *ABA8ox* accumulate more ABA than transgenic *Arabidopsis* plants overexpressing ABA biosynthetic genes [78]. Under salt stress conditions, ABA concentrations in rice are significantly higher, and the expression of ABA biosynthesis genes, including *OsABA2* and *OsAAO3*, is induced [55]. Genes involved in regulating ABA levels are therefore of interest as targets for the development of salt-tolerant crops. Moreover, ion homeostasis in the cell is essential for tolerance to salt stress, and high-affinity potassium transporters (HKTs), Na+/H+ exchangers (NHXs), and salt overly sensitive (SOS) have all been shown to regulate salt stress tolerance in the plant through the balance of intracellular sodium levels [79,80]. Knowledge of these hormone synthesis pathways and salt stress signaling pathways has guided the utilization of plant genetic engineering tools for improving salt stress resistance in crops.

## 4. Enhancing Gene Expression in Crops by Editing uORFs Using the CRISPR System as an Alternative to GM Approaches

Over the last few decades, GM crops have been developed based on the functional study of individual genes. However, the average time and cost for developing GM crops is 13 years and $130 million, respectively, and in many countries, regulatory obstacles still block the commercial cultivation of GM crops [81]. Recently, GE technology, which involves the precise modification of the target genome locus, has enabled the development of bio-engineered crops at a lower cost and in a shorter time period [82,83]. However, this approach has inherent limitations in terms of target gene selection compared to GM technology. GM technology was applied for both overexpression and repression of the target genes. However, GE technology has mainly been applied so far for generating the mutants of target genes by mutation and deletion. Here, we highlight the potential for editing uORFs using GE technology to enhance the expression of target genes associated with various crop traits, thereby expanding the scope of target gene selection and allowing for precise control of target gene expression. For example, editing of the uORF of *LsGGP2*, which encodes an enzyme involved in vitamin C biosynthesis, using the CRISPR-Cas9 system conferred enhanced oxidative stress tolerance and elevated ascorbate levels [84]. In addition, editing the conserved uORF of FvebZIP1.1, which fine-tunes carbon–nitrogen metabolism, resulted in enhanced sugar content in strawberries compared to wild-type [85]

Generally, elevated expression of stress-related genes inhibits plant growth due to the tradeoff between growth and defense. For example, overexpression of *AtJMT* in rice led to drought tolerance but decreased growth and yield [86]. Many salt stress-responsive genes, including *HAK1*, *NHX1*, *SOS1*, *IDS1*, *NAC6*, *EIL2*, *bHLH035*, *JAZ9*, *RSS3*, *ACA6,* and *ABA2* (Table 1), have been identified and functionally characterized by transgenic research. The editing of uORFs in crops using GE technology to target such genes with a verified function associated with stress resistance would be expected to save time and incur less financial cost than traditional breeding or GM approaches, and this strategy is of growing interest [87].

We summarize the process of gene editing of uORFs with the CRISPR system for the development of GE crops (Figure 4): (1) screening mutants or genes for improving the crop traits of interest, such as salt stress-resistance; (2) investigating the gene function and traits, then isolating the candidate genes; (3) analysis and prediction of uORFs in the 5’UTR of the selected gene; (4) investigating the predicted uORFs using a dual-luciferase assay in protoplasts; (5) constructing a CRISPR vector for targeting uORFs and generating the uORF mutant; (6) assessing altered expression of the target gene by uORF mutation and selecting the homozygous mutants without transgene by genotyping; and (7) characterizing the GE crop traits under field conditions.

## 5. Future Prospects

A second ‘green revolution’ is required to support the growing global population and to mitigate negative environmental impacts. To develop biotech crops, it is necessary to identify genes’ underlying quality traits involved in plant defense and growth. The function of stress-responsive genes has been long been characterized in plants using transgenic plant technologies, and recent GE technologies, such as CRISPR and TALEN, are becoming increasingly widespread. The application of uORF mutations via GE has great potential to expand the diversity and global market of GE crops, and will likely be an important factor in stabilizing the food supply and supporting future sustainable agricultural systems.

## Figures and Tables

**Figure 1 ijms-22-03743-f001:**
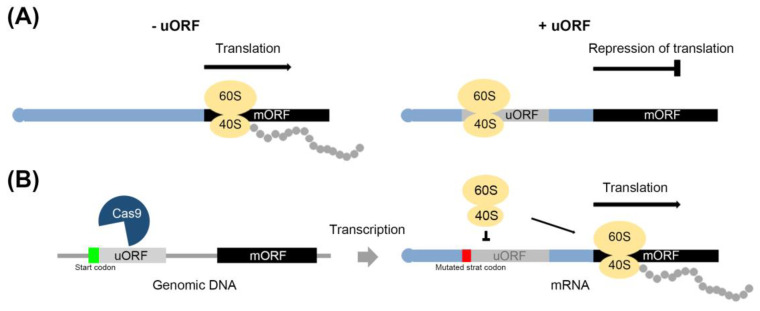
Overview of CRISPR-Cas9-mediated regulation of the repression of translation, as discussed in this review. (**A**) The mRNA (black rectangle) with uORF (gray rectangle) induces ribosome (yellow ovals) stalling in the uORF, which represses translation of the main ORF (mORF). Polypeptide: gray circle. (**B**) Mutation of the start codon region (green or red rectangle) in uORF using CRISPR-Cas9 inhibits ribosome stalling, leading to induced translation of mORF.

**Figure 2 ijms-22-03743-f002:**
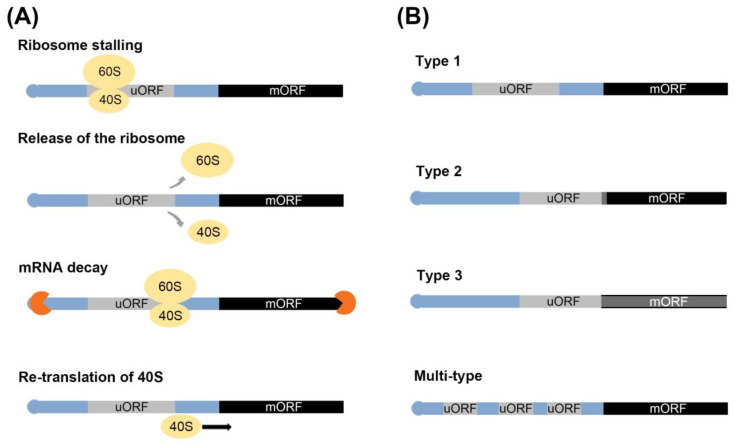
Regulation of uORF-mediated translation and types of uORF. (**A**) The various mechanism of translational repression. (**B**) Types of uORFs. uORFs are classified multiple type and type 1–3. Blue, gray and black rectangles indicate regions of the mRNA, yellow ovals indicate the ribosome complex, and orange circles indicate the exosome.

**Figure 3 ijms-22-03743-f003:**
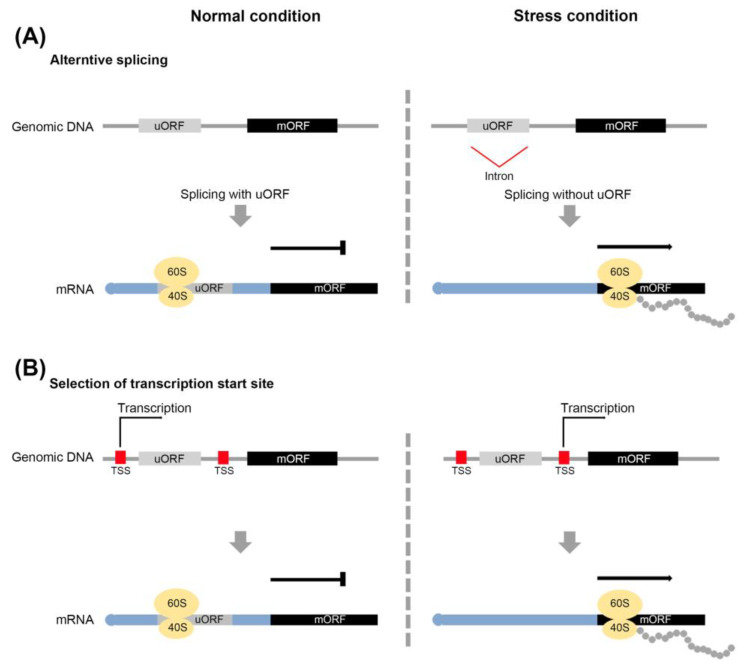
Mechanism of avoiding uORF-mediated repression of gene expression. Alternative splicing (**A**) and alternative transcription start site (TSS) (**B**) to exclude a uORF from mRNA.

**Figure 4 ijms-22-03743-f004:**
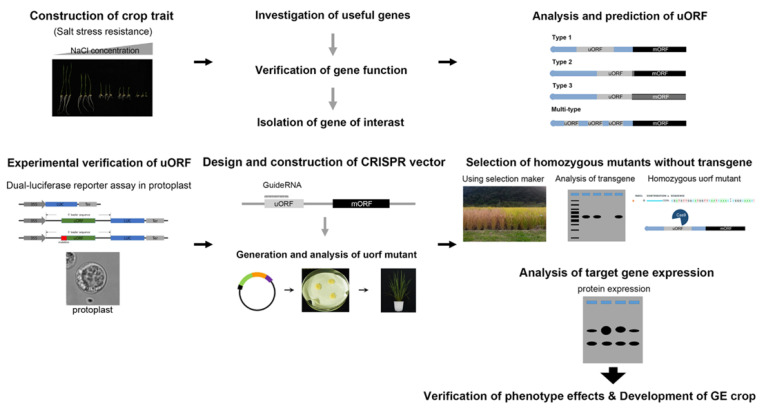
Overview of the process of gene editing of uORFs with the CRISPR system for the development of GE crops.

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
