# Peer review of "Application of Upstream Open Reading Frames (uORFs) Editing for the Development of Stress-Tolerant Crops"

_ijms, 2021, doi:10.3390/ijms22073743_

Round 1

Reviewer 1 Report

Before publishing the manuscript needs extensive English proofreading and editing. Some paragraphs (refer to attached PDF) are describing basic biology concepts in a convoluted way making difficult the understanding of the topic from the reader. there is redundancy in multiple area of the manuscript. The authors should consider adding a section about other methods to regulate gene expression more than a section on development of the first GM crops. Also I couldn't either in the digital version or the printed copy see Figure 4. 

Author Response

Reviewer 1

Before publishing the manuscript needs extensive English proofreading and editing. Some paragraphs (refer to attached PDF) are describing basic biology concepts in a convoluted way making difficult the understanding of the topic from the reader. there is redundancy in multiple area of the manuscript. The authors should consider adding a section about other methods to regulate gene expression more than a section on development of the first GM crops. Also I couldn't either in the digital version or the printed copy see Figure 4. 

A> We revised the previous manuscript as reviewer’s suggestion and removed the redundancy in the manuscript. Also, we replaced figure 4.

Reviewer 2 Report

In this article, the authors review the function of uORFs and the associated mechanisms on gene regulation. They highlight the potential of targeting these regulatory elements for engineering crops that are more resilient to salt stress.

I enjoyed reading this review, it is concise, well written and gives key information on the strategy to follow to target uORFs in rice.

Specific points could be adressed to improve their engineering approach though. Adding more details on the advantages and the drawbacks of targeting uORFs compared to GM approach would add nuance to the strategy used.

Major comments :

1) The authors present a list of salt responsive genes that carry uORFs. Some of them are upregulated while some others are downregulated under salt stress. On line 142-144, the authors only discuss about the genes that are upregulated under salt stress. What would be the impact of mutating uORFs for downregulated genes ?

I am also not sure that all salt regulated genes would be interesting targets for designing salt resilient crops, would it not be more efficient to target key regulators such as transcription factors (e.g OsJAZ9)?

It would be good to clarify these point to refine their engineering strategy.

2) On line 234, the authors mention the limitation of GE crops. While they mostly mention the potential of targeting uORFs, there is no information on the limitations of the system.

For example how much can the expression of target genes be upregulated compared to GM strategies? Would it be sufficient to obtain a stress resilient crop ?

On the other hand, they also do not mention one advantage of targeting uORF with tools such as base editors to fine-tune traits (see Xing et al 2020, https://doi.org/10.1186/s13059-020-02146-5).

Minor comments :

- Line 20 : « The inter-related phenomena of » : can be erased

- Line 58 : The authors take the example of the USDA approved mushroom. It would also be good to replace or include one plant crop

- Line 65 : « have an adverse effect » replace « by negatively regulates »

- Figure 1 panel B : Typo on mutated start codon

- Figure 2a : it would be good to number the different parts of the process, 1) ribosome stalling 2) release of ribosome… or indicate the process by an arrow from first to last step

- Line 129 : what are these 107 genes identified ? The authors could include the list in supplemental data

- Line 166-168 : General information and redundant with line 46-49. I would delete to improve readability

- Line 202 : Replace « new tools » by « new targets to engineer / new genes to target »

- Line 256 : « for targeting uORFs » instead of « for target uORF mutation »

-Figure 4 : remove NGG from the gRNA and replace by PAM site on the uORF DNA. NGG are not included on the gRNA during design and multiple nucleases with different targeting space can be used.

Author Response

Reviewer 2

In this article, the authors review the function of uORFs and the associated mechanisms on gene regulation. They highlight the potential of targeting these regulatory elements for engineering crops that are more resilient to salt stress.

I enjoyed reading this review, it is concise, well written and gives key information on the strategy to follow to target uORFs in rice.

Specific points could be adressed to improve their engineering approach though. Adding more details on the advantages and the drawbacks of targeting uORFs compared to GM approach would add nuance to the strategy used.

Major comments :

1) The authors present a list of salt responsive genes that carry uORFs. Some of them are upregulated while some others are downregulated under salt stress. On line 142-144, the authors only discuss about the genes that are upregulated under salt stress. What would be the impact of mutating uORFs for downregulated genes ?

A> So far, it has been reported that the majority of uORFs function as a negative regulator in the expression of downstream mORFs. Therefore, most of the studies are focused on up-regulation of target mORF by mutation or deletion of uORFs. However, a recent report showed that some uORF functions as a positive regulator for its target gene expression (Lin et al., 2019), and it is expected that target gene expression can be repressed by uORF mutation. On the other hand, we can generate a new uORF in 5’UTR region by the base-changing method. In general, down-regulated genes upon stress play a negative regulator in the stress tolerance mechanism. Therefore, the salt-tolerant crop can be generated by repressing the down-regulated genes by mutation of enhancing uORFs or introducing new uORFs in the target gene transcript.

Ref) Lin, Y.; May, G.E.; Kready, H.; Nazzaro, L.; Mao, M.; Spealman, P.; Creeger, Y.; McManus, C.J. Impacts of uORF codon identity and position on translation regulation. Nucleic Acids Research 2019, 47, 9358-9367, doi:10.1093/nar/gkz681.

I am also not sure that all salt regulated genes would be interesting targets for designing salt resilient crops, would it not be more efficient to target key regulators such as transcription factors (e.g OsJAZ9)?

It would be good to clarify these point to refine their engineering strategy.

A> We screened the target genes in rice by investigating the literature. Parts of candidate genes have already been identified as salt-stress regulators by transgenic research. Other candidate genes also have been expected as important regulators in salt-stress response in the literature. uORFs function as both transcriptional and translational regulators, and we selected them to apply the uORF mutation. We also described the function of candidate genes in salt stress response at section "3.2. Hormone regulation in plant salt stress response"

2) On line 234, the authors mention the limitation of GE crops. While they mostly mention the potential of targeting uORFs, there is no information on the limitations of the system.

For example how much can the expression of target genes be upregulated compared to GM strategies? Would it be sufficient to obtain a stress resilient crop ?

On the other hand, they also do not mention one advantage of targeting uORF with tools such as base editors to fine-tune traits (see Xing et al 2020, https://doi.org/10.1186/s13059-020-02146-5).

 A> We modified the paragraph and inserted the example as reviewer’s suggestion (Line 244-254).

Minor comments :

- Line 20 : « The inter-related phenomena of » : can be erased

A> We remove it as your suggestion. Thank you.

- Line 58 : The authors take the example of the USDA approved mushroom. It would also be good to replace or include one plant crop

A> We replaced the example with GE soybean based on TALEN instead of GE mushroom.

- Line 65 : « have an adverse effect » replace « by negatively regulates »

A> We replace it as your suggestion. Thank you.

- Figure 1 panel B : Typo on mutated start codon

A> We corrected the typo in Fig 1. Thank you.

- Figure 2a : it would be good to number the different parts of the process, 1) ribosome stalling 2) release of ribosome… or indicate the process by an arrow from first to last step

A> We added the arrows to indicate the process.

- Line 129 : what are these 107 genes identified ? The authors could include the list in supplemental data

A> We provide the list of screened genes in Supplemental data.

- Line 166-168 : General information and redundant with line 46-49. I would delete to improve readability

A> We delete the sentence (Line 173-175).

- Line 202 : Replace « new tools » by « new targets to engineer / new genes to target »

A> Thanks for your suggestion. We revised the sentence as your suggestion.

- Line 256 : « for targeting uORFs » instead of « for target uORF mutation »

A> Thanks for your suggestion. We revised the sentence as your suggestion.

- Figure 4 : remove NGG from the gRNA and replace by PAM site on the uORF DNA. NGG are not included on the gRNA during design and multiple nucleases with different targeting space can be used.

A> We removed NGG from the gRNA in Fig 4

Round 2

Reviewer 1 Report

The authors submitted a revised version of the manuscript that disregarded multiple suggestions form this reviewer. I am not sure if the authors noticed the comments in the reviewed PDF file attached to the first review. In order to avoid any misunderstanding I am summarizing below my suggestions

1) Line 22-23 and line 53

the arguments that gene edited crops have been developed to overcome the regulatory constrains of the GM crops needs to be further elaborated (provide references and examples) since currently GE trait product developments are suffering from the same constrains as the GM ones

2) Line 25 and Line 60

Reference needed for the statements. what do the authors mean with growing rapidly? To my knowledge the cultivation and use of GE crops is still limited

3) Line 85

English proofing for the following sentence: "this process can (be) divided in three stages

4) Line 119

English proofing

Rice is an excellent model to study the significance of uORFs in stress related gene expression (since) its genome has been extensively  

5) Figure 3 

Alterntive splicing is spelled wrong 

6) Line 174

Consistency either each crop listed has the latin botanical classification or remove Gossypium hirsutum

7) Line 186

old reference please use a more recent one from either the same source or a different one

--------------------------

The description of mRNA is not well organized and the same is true for the one about the avoidance of the uORF-mediated repression of gene expression during stress response. Section 3 and 3.1 are unnecessarily lengthy. 

Author Response

Reviewer 1

The authors submitted a revised version of the manuscript that disregarded multiple suggestions form this reviewer. I am not sure if the authors noticed the comments in the reviewed PDF file attached to the first review. In order to avoid any misunderstanding I am summarizing below my suggestions

A> We did not recognize attached pdf file before and we are very sorry about that. We appreciate for your detailed suggestions and those are greatly helpful to us.

1) Line 22-23 and line 53

the arguments that gene edited crops have been developed to overcome the regulatory constrains of the GM crops needs to be further elaborated (provide references and examples) since currently GE trait product developments are suffering from the same constrains as the GM ones

A> As you mentioned, there are also many constrains for application of GE crops now. But it is expected that regulations will be loosened because of the benefits of GE technique. We modified the sentences and added the references (Line 59-62). We also modified the sentences in the Abstract.

2) Line 25 and Line 60

Reference needed for the statements. what do the authors mean with growing rapidly? To my knowledge the cultivation and use of GE crops is still limited

A> As you mentioned, there are still many constrains for application of GE crops now. We modified the sentences (Line 59-62)

3) Line 85

English proofing for the following sentence: "this process can (be) divided in three stages

A> We revised the sentence as your suggestion. Thank you

4) Line 119

English proofing

Rice is an excellent model to study the significance of uORFs in stress related gene expression (since) its genome has been extensively  

A> We revised the sentence as your suggestion. Thank you

5) Figure 3 

Alterntive splicing is spelled wrong

A> We corrected the typo.

6) Line 174

Consistency either each crop listed has the latin botanical classification or remove Gossypium hirsutum

A> We removed the botanical classification for consistency.

7) Line 186

old reference please use a more recent one from either the same source or a different one

A> We updated the information with 2019 ISAAA report.

--------------------------

The description of mRNA is not well organized and the same is true for the one about the avoidance of the uORF-mediated repression of gene expression during stress response. Section 3 and 3.1 are unnecessarily lengthy. 

A>We added the description of mRNA in line 65, and actually it has not been identified the detailed mechanism of leaky scanning of uORF under stress condition. We described story of GM crop development to connect the story of GE crop development (Section 4)

Round 3

Reviewer 1 Report

The authors addressed most of my comments.

  • Please note that I would consider changing the sentence in line 59-63

In contrast to the previous GM techniques which randomly introduce undirected (randomly already implies that there is no control in the insertion of your change) changes, GE allows the precise modification and it is identical to those derived from conventional breeding [11, 12] Thus, it is anticipated that many GE crops based on CRISPR (there is an example of TALENS generated crop in line 57 so I would rather use a more inclusive terminology like GE crops) can be a more acceptable product  than the alternatives to GM crops [13].

  • line 200

improve the traits

  • line 201

Thus, elucidation of molecular mechanisms of hormone biosynthesis and signaling pathways are providing new genes to target for developing GM crops Do the authors mean GE crops?

Author Response

Reviewer 1

A> First of all, we are grateful to your valuable comments, which have allowed us to significantly improve the quality and impact of our manuscript. We have taken all of your comments into consideration and revised manuscript.

1) Please note that I would consider changing the sentence in line 59-63

In contrast to the previous GM techniques which randomly introduce undirected (randomly already implies that there is no control in the insertion of your change) changes, GE allows the precise modification and it is identical to those derived from conventional breeding [11, 12] Thus, it is anticipated that many GE crops based on CRISPR (there is an example of TALENS generated crop in line 57 so I would rather use a more inclusive terminology like GE crops) can be a more acceptable product than the alternatives to GM crops [13].

A> We appreciate your comment. We modified the sentences (Line 59-63)

2) line 200

improve the traits

A> We revised the sentence as your suggestion (Line 200). Thank you

3) line 201

Thus, elucidation of molecular mechanisms of hormone biosynthesis and signaling pathways are providing new genes to target for developing GM crops Do the authors mean GE crops?

A> We heartily appreciate your comments. We are sorry for this. These errors have been corrected in the revised manuscript (line 203).